# Iron Dysregulation Signature in Pediatric Leukemia: In-Depth Biomarkers of Iron Metabolism Involving Matriptase-2 and Neogenin-1

**DOI:** 10.3390/cancers17152495

**Published:** 2025-07-29

**Authors:** Monika Łęcka, Artur Słomka, Katarzyna Albrecht, Michał Romiszewski, Jan Styczyński

**Affiliations:** 1Department of Pediatric Hematology and Oncology, Jurasz University Hospital, Collegium Medicum Nicolaus Copernicus University Torun, 85-094 Bydgoszcz, Poland; monika.lecka@cm.umk.pl; 2Department of Pathophysiology, Collegium Medicum Nicolaus Copernicus University Torun, 85-094 Bydgoszcz, Poland; artur.slomka@cm.umk.pl; 3Department of Digital Medicine, Implementation & Innovation, National Medical Institute of the Ministry of Interior and Administration, 02-507 Warsaw, Poland; 4Department of Oncology, Hematology, Bone Marrow Transplantation and Pediatrics, Medical University of Warsaw, 02-091 Warsaw, Poland; katarzyna.albrecht@wum.edu.pl (K.A.); michal.romiszewski@uckwum.pl (M.R.)

**Keywords:** iron, hemojuvelin, leukemia, hematopoietic cell transplantation

## Abstract

In our study, we investigated how children with acute leukemia (AL) manage iron levels in their bodies during different stages of treatment. We focused on specific proteins that help regulate hepcidin, a hormone that controls how the body absorbs and stores iron. We identified a potential mechanism that may interfere with this regulation, especially in children who receive multiple blood transfusions. Although we studied several blood-based markers related to this process, none of them proved useful in predicting how severe the disease would be. Further research is needed to evaluate whether these markers could have clinical applications in the future.

## 1. Introduction

This year marks the 25th anniversary of the identification of hepcidin, a protein playing a fundamental role in human iron homeostasis [1,2]. This seminal discovery, subsequently reinforced by pivotal studies describing hepatic synthesis [3] and elucidating the mechanism of action [4], inaugurated a new era in iron metabolism research, casting novel insights into the pathogenesis of various disorders, particularly anemia and iron overload [5,6,7].

The principal mechanism of hepcidin involves the blockade of ferroportin, a transporter protein facilitating iron export from cells—primarily enterocytes, various hepatic cells, and erythroblast precursors—into the circulation, thus protecting the organism from the toxic effects of excessive iron accumulation [8,9]. Subsequent intensive research into iron metabolism has elucidated a sophisticated regulatory network governing systemic iron stores, as well as the intricate, multilayered mechanisms modulating hepcidin concentrations [10,11].

Critically significant in this context is hemojuvelin (HJV), a key component of the BMP/SMAD/HJV molecular signaling pathway, exhibiting dual functionality with respect to hepcidin regulation [12,13,14]. Membrane-bound HJV, expressed on hepatocytes, enhances hepcidin synthesis, a mechanism directly contrasted by soluble HJV (sHJV), which decreases hepcidin expression [15]. These findings have intensified research focus on HJV and its circulating form, without, however, concluding explorations into their roles in iron metabolism. Subsequent investigations have elucidated that matriptase-2 (TMPRSS6) and neogenin-1 (NEO1) act as pivotal regulators of hemojuvelin (HJV) expression, constituting a complex molecular axis wherein TMPRSS6 mediates the downregulation of membrane-bound HJV [16,17], a process further facilitated by NEO1 [18]. Consequently, the HJV-TMPRSS6-NEO1 axis emerges as cardinal in maintaining iron homeostasis through precise control of hepcidin expression.

All these intricate molecular mechanisms primarily serve to protect biomolecules and cellular structures from iron-induced toxicity [19]. It is thus unsurprising that research into iron-overload conditions actively seeks biomarkers to evaluate this phenomenon and predict clinical outcomes or therapeutic responses [20]. Our research group has undertaken similar efforts among pediatric patients with acute leukemias, demonstrating that disturbances in systemic iron homeostasis are strongly associated with therapy type and that sHJV potentially predicts unfavorable outcomes in these children [21,22]. Despite their pioneering nature, these studies require further enhancement through additional laboratory parameters exhibiting significant correlations with sHJV and hepcidin.

Consequently, we designed the present study to meticulously explore the interrelationships between proteins governing sHJV concentrations, clinical status, and applied therapeutic modalities. Again, we focused on pediatric acute leukemia patients to achieve deeper insights into the mechanisms underpinning iron metabolism disorders in this vulnerable population. Our attention specifically encompassed the HJV-TMPRSS6-NEO1 axis, providing a broader understanding of iron metabolism disturbances, which frequently predispose children with acute leukemias to iron overload and adverse clinical outcomes [23,24].

Through rigorous patient selection, we assessed serum concentrations of HJV, TMPRSS6, and NEO1 in relation to treatments such as chemotherapy and hematopoietic cell transplantation (HCT). This study robustly confirmed the role of sHJV as a potential predictor of mortality in pediatric acute leukemias. Nonetheless, further evaluation of TMPRSS6 and NEO1 remains essential to determine their prognostic utility for overall survival (OS) and event-free survival (EFS). Understanding these molecular intricacies is imperative for advancing precise diagnostic tools and personalized therapeutic strategies, ultimately improving survival outcomes in pediatric acute leukemia patients.

## 2. Materials and Methods

The study cohort comprised a diverse pediatric population enrolled between 2021 and 2024 at two academic tertiary care institutions in Poland: the Department of Pediatrics, Hematology, and Oncology at University Hospital No. 1 in Bydgoszcz, Collegium Medicum of Nicolaus Copernicus University in Torun, and the Department of Pediatric Oncology, Hematology, Clinical Transplantology, and Pediatrics at the Medical University of Warsaw.

The recruited population included both hematologically healthy children, evaluated during scheduled outpatient visits, as well as pediatric patients at various stages of acute leukemia (AL). Among the latter, subgroups encompassed children at the time of initial diagnosis prior to the commencement of any antineoplastic therapy, patients who had completed intensive chemotherapy, and those who had undergone hematopoietic cell transplantation (HCT).

Children diagnosed with acute lymphoblastic leukemia (ALL) were managed according to the AIEOP-BFM-ALL-2017 protocol, whereas those with acute myeloid leukemia (AML) were treated following the AML-BFM-2019 protocol. In patients who underwent HCT, therapeutic management was based on the corresponding disease-specific chemotherapy protocols, followed by conditioning therapy prior to transplantation. In cases of non-malignant hematologic indications, supportive and immunosuppressive regimens were applied as per institutional standards.

Peripheral blood samples were collected under standardized conditions, with the timing of phlebotomy adapted to the clinical status of each study subgroup. In healthy pediatric individuals, venous blood was obtained during scheduled outpatient visits to university-affiliated pediatric clinics. In children newly diagnosed with AL, sample collection was performed at the time of diagnosis, prior to the initiation of any antineoplastic therapy. In patients who had completed intensive chemotherapy or undergone HCT, blood was drawn one month following the completion of the respective therapeutic protocol during routine post-treatment follow-up.

All specimens were obtained via peripheral venipuncture following an overnight fasting period, using serum-separating tubes (Becton Dickinson, Franklin Lakes, NJ, USA). After collection, samples were maintained at ambient temperature for a minimum of 30 min to allow complete coagulation. Subsequently, samples were centrifuged at 2000× *g* for 20 min at room temperature to facilitate serum separation. The resultant supernatant (serum fraction) was carefully aspirated, aliquoted into cryogenic storage vials, and immediately stored at −80 °C pending biochemical analysis. Samples exhibiting any degree of hemolysis were excluded from further evaluation to preserve analytical accuracy and pre-analytical integrity.

Quantitative assessment of serum concentrations of selected biomarkers of iron metabolism was performed using enzyme-linked immunosorbent assay (ELISA) methodology. Serum concentrations of TMPRSS6 were measured using the Human TMPRSS6 (Sandwich ELISA) Kit (LS-F34191), characterized by excellent analytical precision, with intra-assay and inter-assay coefficients of variation (CV) below 8% and 10%, respectively, and a detection sensitivity of 0.075 ng/mL. Serum concentrations of NEO1 were determined using the Human NGN/Neogenin (Sandwich ELISA) Kit (LS-F38105), exhibiting intra-assay and inter-assay CVs below 10%, with a sensitivity of 0.094 ng/mL. Both kits were supplied by LSBio (Vector Laboratories, Inc., Newark, CA, USA). Serum concentrations of soluble hemojuvelin (sHJV) were analyzed using the ELISA Kit for Hemojuvelin (HJV, CEB995Hu) from Cloud-Clone Corp. (Katy, TX, USA), with intra-assay CV < 10%, inter-assay CV < 12%, and a sensitivity threshold of 4.93 ng/mL.

### Statistical Analysis

Statistical analyses were conducted with SPSS version 29 (IBM Corp., Armonk, NY, USA). Continuous variables were assessed with the Wilcoxon rank-sum test when more than two groups were compared and with the Mann–Whitney *U* test for two-group comparisons. Categorical variables were evaluated with the chi-square test or, when expected counts were insufficient, the Fisher exact test. Correlations among quantitative parameters were examined using Spearman rank correlation. The primary objective of this study was to evaluate the concentrations of selected parameters associated with iron metabolism in pediatric acute leukemia with respect to the therapeutic stage and relevant biological variables, including the cumulative volume of transfused red blood cell concentrates, baseline iron stores, and the extent of systemic inflammation. As an exploratory analysis, overall survival (OS) and event-free survival (EFS) were also assessed. Both endpoints were measured from the enrollment date, defined as the day on which iron metabolism parameters were obtained. OS was defined as the interval from enrollment to death from any cause or to the last follow-up, while EFS referred to the period without relapse or disease progression; relapse was defined as more than five percent bone marrow blasts or reappearance of the underlying disease. Survival curves were estimated using the Kaplan–Meier method, and differences between groups were tested using the log-rank test. All statistical analyses were performed using two-tailed tests, with a *p*-value of less than 0.05 considered indicative of statistical significance. In cases involving multiple comparisons conducted on the same dataset, statistical significance was determined using the Bonferroni-adjusted alpha threshold to control for type I error inflation.

## 3. Results

### 3.1. Clinical and Demographic Profile of Patients

Owing to the collaborative engagement of two tertiary pediatric hematology and oncology centers based in Bydgoszcz and Warsaw, it was feasible to recruit a relatively substantial cohort of pediatric patients (*n* = 149), comprising 76 girls and 73 boys, with a median age of 8 years (interquartile range [IQR]: 6–12.5 years). The entire cohort was stratified into four clinically defined subpopulations based on the therapeutic interventions administered.

The first subgroup consisted of 19 healthy pediatric individuals (median age: 10 years; IQR: 9–13 years), serving as the control population. The second subgroup encompassed 43 children, newly diagnosed with acute leukemia—specifically acute lymphoblastic leukemia (ALL; *n* = 40) and acute myeloid leukemia (AML; *n* = 3)—prior to the initiation of any antineoplastic therapy (median age: 8 years; IQR: 6–14 years). The third subgroup (*n* = 55) included patients who had undergone intensive chemotherapy for ALL (*n* = 52) or AML (*n* = 3), with a median age of 8.7 years (IQR: 6–11 years). The fourth and final subgroup comprised 32 children who had undergone hematopoietic cell transplantation (HCT) for ALL (*n* = 8), AML (*n* = 14), or other malignancies qualifying for HCT (*n* = 10), with a median age of 8 years (IQR: 4–14 years). Notably, all patients in this subgroup were post-HCT. This subgroup also comprised patients with other malignancies qualifying for HCT, including conditions such as severe aplastic anemia (SAA, *n* = 3), neuroblastoma (NBL, *n* = 3), severe congenital neutropenia (SCN, *n* = 1), myelodysplastic syndrome (MDS, *n* = 1), Ewing sarcoma (ES, *n* = 1) and anaplastic large B-cell lymphoma (ALCL, *n* = 1). A total of 27 patients underwent allogeneic HCT, comprising 5 cases with related donors and 22 with matched unrelated donors; in addition, autologous HCT was performed in 5 individuals.

A total of 122 individuals within the cohort received transfusions of packed red blood cells (PRBCs), representing the vast majority of pediatric patients diagnosed with malignancies, as opposed to healthy controls who did not require transfusional support. The median number of transfused PRBC units was 7 (IQR: 1–14), exhibiting a progressive increase proportional to the therapeutic burden, rising from a median of 1 unit (IQR: 0.6–2) in newly diagnosed patients to 10 units (IQR: 7–16) post-chemotherapy and 23 units (IQR: 12–34) in the post-HCT group. These differences in transfusion burden were statistically significant. No statistically significant disparities in age or sex distribution were observed between the subgroups. The comprehensive demographic and clinical data are compiled in Table 1.

### 3.2. Biomarker Trajectories Along the Continuum of Pediatric Leukemia Therapy

In the subsequent analytical phase, intergroup comparisons were conducted to evaluate serum concentrations of three selected biomarkers: sHJV, TMPRSS6, and NEO1. Distinct and biologically compelling alterations in the concentrations of TMPRSS6 and NEO1 were discerned. TMPRSS6 exhibited a stepwise elevation corresponding to the therapeutic trajectory—being lowest in healthy controls and progressively increasing among newly diagnosed patients and post-chemotherapy recipients, and peaking in the post-HCT cohort. Statistically robust differences were identified between healthy individuals and both chemotherapy-treated and HCT-treated subgroups.

An analogous trend was observed for serum NEO1, whose levels similarly escalated in accordance with treatment intensity. Statistically significant differences were also confirmed between the control group and the respective post-treatment cohorts.

In contrast to the aforementioned biomarkers, sHJV demonstrated an inverse association with treatment burden, manifesting a decremental trajectory with the lowest concentrations observed in the HCT subgroup. This inverse pattern further substantiates our earlier observations [22]. The detailed results of the biomarker analyses are presented in Table 2.

### 3.3. Iron Status and Transfusion-Related Changes in TMPRSS6, NEO1, and sHJV

In a subsequent phase of analysis, the potential relationship between systemic iron status and serum concentrations of TMPRSS6, NEO1, and sHJV was investigated. Systemic iron availability was defined using serum ferritin concentrations, obtained as part of standard clinical diagnostics conducted in the central laboratories of university hospitals [25].

Following the determination of the median ferritin value, the cohort was stratified into two subgroups. Comparative analysis was then carried out to assess differences in the circulating concentrations of TMPRSS6, NEO1, and sHJV between individuals with ferritin concentrations below versus above the median threshold.

Our analysis confirmed that patients with higher serum ferritin concentrations exhibited significantly elevated levels of TMPRSS6 and NEO1 compared to those classified below the median threshold. Conversely, sHJV concentrations were markedly lower among individuals with increased ferritin levels. These findings are summarized in Table 3.

To further explore the potential association between transfusional burden and the serum concentrations of TMPRSS6, NEO1, and sHJV, an analogous analytical approach was employed. Specifically, patients were stratified into two subgroups according to the median number of packed red blood cell (PRBC) units transfused. Healthy children were excluded from this analysis, as they had not received any transfusions. A highly similar pattern was observed with respect to the number of transfused PRBC units. Specifically, patients who received a greater number of transfusions demonstrated significantly higher serum concentrations of TMPRSS6 and NEO1, as well as lower levels of sHJV, in comparison to those with fewer administered PRBC units. These findings are also presented in Table 3.

### 3.4. Inflammation-Related Changes in TMPRSS6, NEO1, and sHJV

Given the well-established interplay between systemic iron homeostasis and inflammation, this segment of our analysis aimed to evaluate whether serum concentrations of TMPRSS6, NEO1, and sHJV are associated with a widely utilized clinical marker of inflammation—C-reactive protein (CRP) [26]. CRP values were obtained from routinely performed laboratory tests conducted in hospital-based diagnostic laboratories.

Following the determination of the cohort-specific median CRP concentration, the entire group of 149 pediatric patients was stratified into two subgroups according to whether their serum CRP values were below or above this threshold. Regarding CRP, a statistically significant difference was observed exclusively in the serum concentration of sHJV, which was lower among patients with CRP levels exceeding the median. Nonetheless, this association warrants cautious interpretation, as the corresponding *p*-value approached the threshold of statistical significance (Table 3).

### 3.5. Correlation Patterns of TMPRSS6, NEO1, and sHJV

Building upon the current body of evidence that strongly supports the existence of a functional TMPRSS6-NEO1 axis with potential regulatory effects on sHJV, the logical extension of this hypothesis involves a quantitative investigation of interrelationships between these proteins through the assessment of correlations among their circulating concentrations. A Spearman correlation heatmap was constructed to visualize the strength and statistical significance of monotonic relationships between the analyzed variables (Figure 1).

Spearman’s rank correlation coefficients (r) and corresponding *p*-values were determined, revealing several noteworthy associations. Notably, serum sHJV concentrations demonstrated statistically significant inverse correlations not only with TMPRSS6 and NEO1 but also with serum ferritin concentrations, the number of transfused PRBC units, and CRP concentrations.

A significant positive association was observed between serum concentrations of TMPRSS6 and NEO1. Furthermore, both parameters demonstrated a direct correlation with serum ferritin levels and the cumulative number of packed red blood cell (PRBC) units administered throughout the course of treatment.

### 3.6. Discriminative Potential of TMPRSS6, NEO1, and sHJV in Survival Analysis: A Kaplan–Meier Perspective

The subsequent stage of the analysis focused on determining the association between overall survival (OS) and event-free survival (EFS) and the circulating concentrations of TMPRSS6, NEO1, and sHJV in pediatric patients diagnosed with AL who had undergone chemotherapy and HCT. The median follow-up period was 3.0 years in the chemotherapy group and 3.5 years in the HCT cohort. Among patients treated with chemotherapy, 11 deaths and 7 relapses were recorded, whereas 8 deaths and 3 relapses occurred in individuals who underwent HCT. Notably, in 4 of the fatal cases, death was preceded by disease relapse. Survival analyses for OS and EFS were performed using univariate statistical methods, with patients stratified based on median dichotomization of serum concentrations for each respective biomarker (TMPRSS6, NEO1, and sHJV).

Survival analyses were initially performed within the chemotherapy-treated subgroup, as illustrated in Figure 2. No statistically significant associations were identified between serum concentrations of the investigated biomarkers and either overall survival (OS) or event-free survival (EFS). Subsequent evaluation in the HCT subgroup similarly failed to demonstrate any prognostic value of these biomarkers with respect to survival endpoints. The relevant data are depicted in Figure 3.

## 4. Discussion

To our knowledge, this is the first clinical study to characterize the dysregulation of the TMPRSS6-NEO1-HJV axis in the context of transfusional iron overload in pediatric acute leukemia. Our findings demonstrate that increasing iron burden, as evidenced by elevated serum ferritin concentrations and cumulative exposure to packed red blood cell transfusions, is associated with significantly increased levels of TMPRSS6 and NEO1. TMPRSS6, with functional support from NEO1, cleaves membrane-bound hemojuvelin (mHJV), thereby reducing its surface availability. As a result, the hepcidin-stimulatory signal derived from mHJV is diminished, leading to altered regulation of systemic iron homeostasis.

These data support a mechanistic framework in which transfusion-associated iron overload activates the TMPRSS6-NEO1 signaling axis, resulting in the depletion of functional membrane-bound hemojuvelin (mHJV), perturbation of hepcidin regulatory dynamics, and subsequent systemic iron accumulation. This molecular pathway appears to constitute a critical interface between transfusional iron excess and impaired homeostatic feedback in pediatric leukemogenesis. Concurrently, the present findings suggest that elevated iron burden may suppress the bioactive pool of secreted soluble hemojuvelin (sHJV), thereby diminishing its inhibitory influence on hepatic hepcidin transcription. This may facilitate iron-driven upregulation of hepcidin expression and a consequent reduction in circulating iron availability, representing a compensatory mechanism that could serve an adaptive function in the setting of chronic transfusional siderosis in pediatric patients with acute leukemia. In order to support better comprehension, Figure 4 provides a simplified graphical representation of the discussed concept.

Although acute leukemia is a relatively rare pediatric malignancy, we succeeded in assembling a substantial and clinically well-defined cohort through collaboration between two academic hematology–oncology centers. This allowed for a systematic analysis of iron metabolism regulators across different treatment stages and in comparison to healthy children, providing a unique opportunity to explore mechanistic alterations that would be difficult to capture in smaller or less diverse populations. To our knowledge, this is the first study to evaluate TMPRSS6, NEO1, and sHJV in these distinct pediatric subgroups. Our findings delineate a novel iron-regulatory mechanism involving altered mHJV availability due to TMPRSS6-NEO1 activity and reveal how this axis is modulated throughout leukemia progression and treatment. The three analyzed biomarkers (TMPRSS6, NEO1 and sHJV) showed altered serum concentrations in the studied cohort, indicating disturbances in iron regulatory mechanisms. Given their potential involvement in disease biology, further comprehensive studies are urgently needed to assess their prognostic significance for overall survival (OS) and event-free survival (EFS) in pediatric acute leukemia.

TMPRSS6 is a key modulator of iron homeostasis through its regulatory impact on HAMP expression [27]. Prior genomic investigations have strongly implicated this serine protease in the orchestration of iron metabolism, particularly under states of deficiency [28,29,30]. The phenomenon we report in pediatric subjects with acute leukemia, especially following intensive chemotherapeutic protocols, constitutes a novel observation and unequivocally suggests a perturbation at one or more levels of the molecular hierarchy governing iron-regulatory proteins. This is mechanistically paradoxical, as excess systemic iron would typically be expected to suppress TMPRSS6 activity.

Such findings remain exceedingly uncommon, although prior evidence indicates that *TMPRSS6* mRNA may be upregulated in the setting of chronic exogenous iron administration [31]. Nonetheless, these observations were derived from non-human models and did not reflect conditions of extreme iron burden. Of particular interest is the role of NEO1, as it remains an open question whether iron excess per se modulates its expression. Our data demonstrate a potential relationship: elevated NEO1 concentrations were observed in individuals with clinically significant iron overload, with apparent correlations to serum ferritin and cumulative PRBC exposure. Whether NEO1 is mechanistically responsible for the augmented TMPRSS6 activity documented in our cohort, and whether the serum abundance of these proteins correlates with therapeutic responsiveness, remain unresolved. Nevertheless, the present study underscores the need for further interrogation of the TMPRSS6-NEO1 axis in the context of transfusional siderosis in pediatric hematologic malignancies.

Naturally, certain methodological limitations must be acknowledged. We did not assess additional modulatory factors that may influence the TMPRSS6-NEO1-sHJV triad, including highly sensitive and specific inflammatory markers. In particular, the absence of quantification of cytokines such as interleukin-6—and other members of the interleukin family—constitutes a notable limitation, given their well-established role in mediating systemic inflammation and their potential impact on sHJV regulation [32]. Moreover, other proteases, such as furin [33], may participate in sHJV release from the cell membrane. Subsequent analyses should include quantification of such candidates. The survival analysis in this study was exploratory and did not reveal statistically significant associations, which, together with the limited subgroup sizes and clinical heterogeneity, precludes drawing definitive prognostic conclusions. Nonetheless, although alternative statistical methods aimed at prognostic modeling and survival prediction could potentially yield additional insights, their implementation was constrained in our study by methodological limitations inherent to pediatric cohorts, notably the restricted patient sample size. Future research involving larger pediatric populations, expanded longitudinal assessment, and integrative analyses that correlate serum biomarker concentrations with their tissue-specific protein expression profiles would be instrumental in further validating and clarifying the prognostic role of TMPRSS6, NEO1, and sHJV in pediatric acute leukemia, as well as in elucidating their biological contributions to iron dysregulation observed in these patients.

The present study is the first to characterize concurrent dysregulation of TMPRSS6, NEO1, and sHJV in transfusion-driven iron overload among pediatric leukemia patients. Excess systemic iron was associated with a paradoxical up-regulation of TMPRSS6 and NEO1 together with depletion of sHJV, exposing a bidirectional disturbance within the HAMP regulatory axis. These findings warrant prospective validation and mechanistic dissection of the TMPRSS6-NEO1-sHJV pathway, and they suggest that therapeutic modulation of this axis may mitigate iron-related morbidity in childhood hematologic malignancies.

## 5. Conclusions

Taken together, the findings of this investigation elucidate a previously uncharacterized perturbation in the TMPRSS6-NEO1-sHJV molecular axis, implicating it as a critical determinant of systemic iron homeostasis in pediatric acute leukemia. The documented elevation of TMPRSS6 and NEO1 concomitant with reduced sHJV strongly suggests their mechanistic involvement in iron overload, exacerbated by therapeutic and transfusional exposures. The inverse association of sHJV with systemic iron parameters and inflammatory markers further supports its potential utility as a sensitive and clinically informative biomarker for monitoring leukemia-related iron dysregulation. These observations lend credence to the hypothesis that modulation of this axis might conceivably influence the trajectory of iron-related complications in pediatric patients, though such an approach remains to be rigorously substantiated. Future prospective studies should aim not only to assess the prognostic significance of TMPRSS6, NEO1, and sHJV but also to delineate the molecular underpinnings of their interplay and examine their potential as therapeutic targets. A deeper understanding of this regulatory network may ultimately contribute to the refinement of risk stratification algorithms and the development of individualized treatment strategies, thereby enhancing clinical outcomes in pediatric acute leukemia.

## Figures and Tables

**Figure 1 cancers-17-02495-f001:**
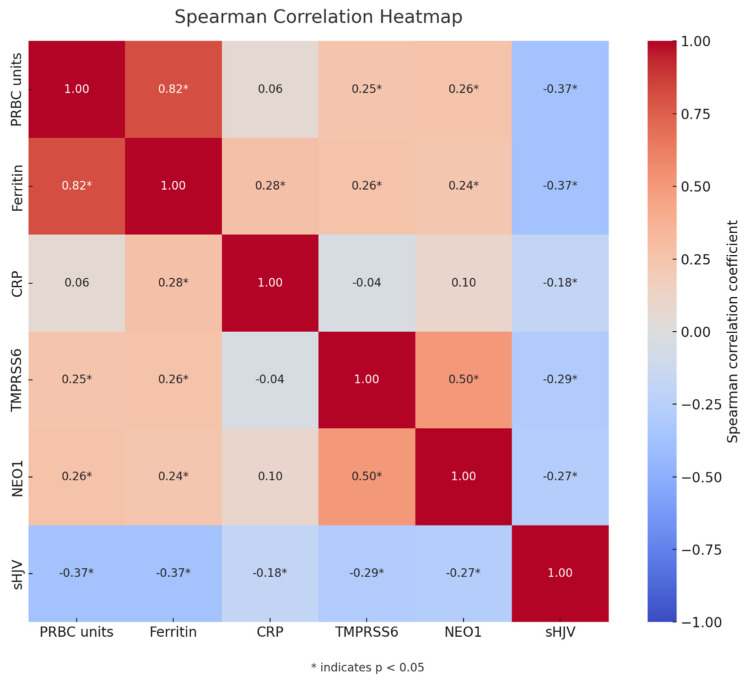
Spearman correlation heatmap. Color intensity reflects the strength and direction of correlation (red: positive, blue: negative). Asterisks indicate statistically significant correlations (*p* < 0.05).

**Figure 2 cancers-17-02495-f002:**
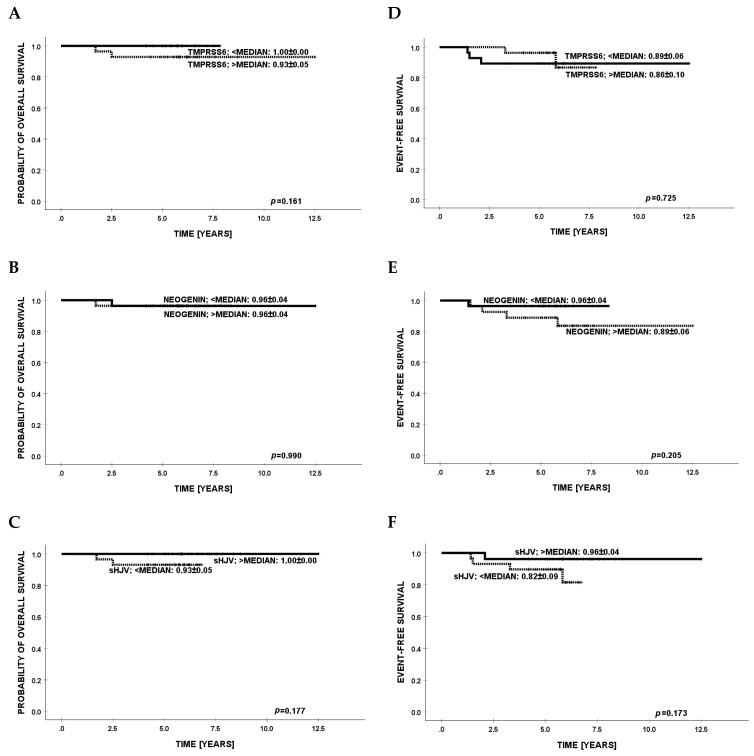
Kaplan–Meier estimates of overall survival (OS; panels **A**–**C**) and event-free survival (EFS; panels **D**–**F**) in pediatric patients after chemotherapy, stratified according to median serum concentrations of TMPRSS6, NEO1, and sHJV.

**Figure 3 cancers-17-02495-f003:**
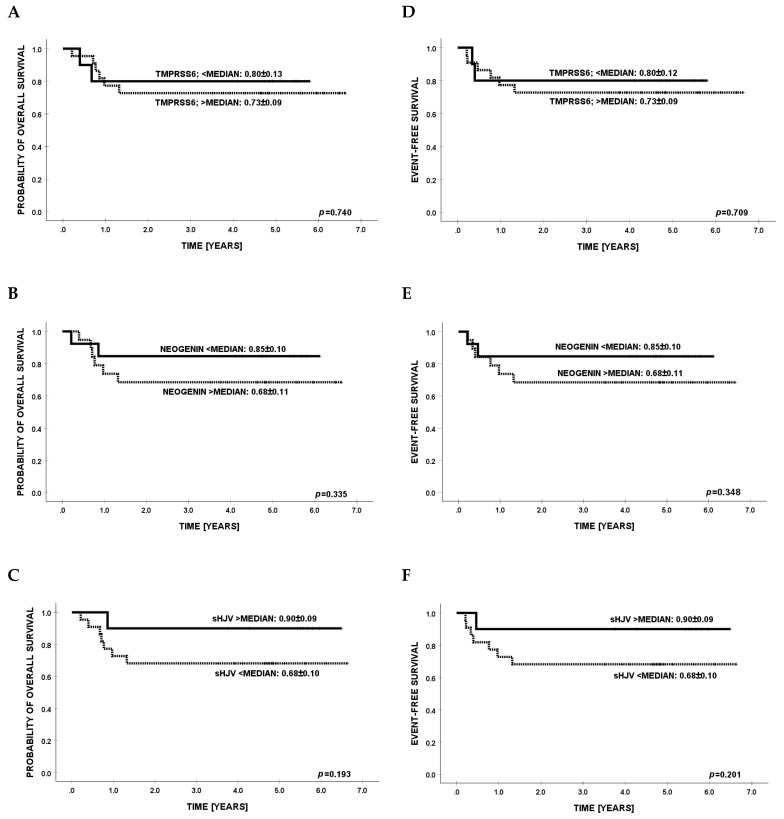
Kaplan–Meier estimates of overall survival (OS; panels (**A**–**C**)) and event-free survival (EFS; panels (**D**–**F**)) in pediatric recipients of hematopoietic cell transplantation (HCT), stratified according to median serum concentrations of TMPRSS6, NEO1, and sHJV.

**Figure 4 cancers-17-02495-f004:**
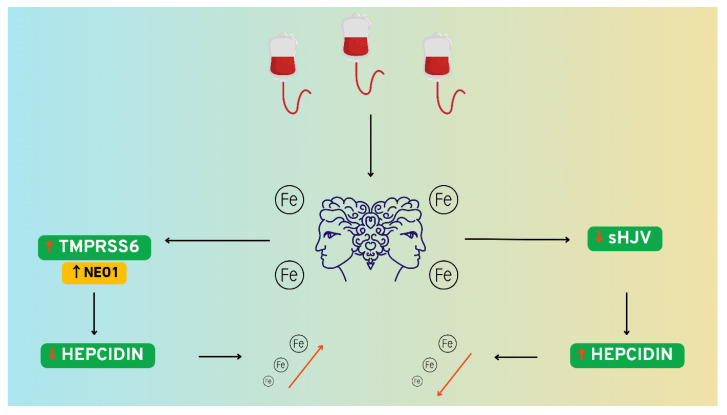
Hypothetical mechanistic framework underlying our observations and its potential ramifications for systemic iron homeostasis. The findings of our investigation reveal that transfusion-induced iron overload may exert a dichotomous pathophysiological effect. On one side of this axis, through a relatively well-elucidated molecular mechanism, the supraphysiological accumulation of iron leads to a diminution in circulating concentrations of soluble hemojuvelin (sHJV). This reduction is associated with the subsequent upregulation of *HAMP* gene expression, culminating in elevated hepcidin synthesis. The latter, in turn, inhibits both the mobilization of iron from physiological stores and its intestinal absorption, primarily at the level of the duodenum. The distinctive contribution of our study lies in the identification of a parallel and mechanistically antagonistic phenomenon—namely, a transfusion-associated increase in serum levels of transmembrane serine protease 6 (TMPRSS6), plausibly potentiated by the co-activity of neogenin-1 (NEO1). This upregulation appears to facilitate the proteolytic cleavage of membrane-bound hemojuvelin (mHJV), resulting in the downregulation of its expression. As a direct consequence, the biosynthesis of hepcidin is attenuated, a process that may predispose to or amplify subsequent systemic iron overload. The figure was created using the Canva graphic design platform (Canva Pty Ltd., Sydney, Australia).

**Table 1 cancers-17-02495-t001:** Demographic and transfusion characteristics of pediatric subjects enrolled in the study.

Clinical Subpopulation	Median Age (Years, IQR)	Sex Composition	ALL (*n*)	AML (*n*)	Other Malignancies (*n*)	HCT (*n*)	Patients Receiving PRBC (*n*)	PRBC Units, Median (IQR)
Healthy pediatric group (*n* = 19)	10.0 (9.0–13.0)	8 boys, 11 girls	0	0	0	0	0	0.0 (0.0–0.0)
Newly diagnosed patients (*n* = 43)	8.0 (6.0–14.0)	20 boys, 23 girls	40	3	0	0	35	1.0 (0.6–2.0)
Patients following chemotherapy (*n* = 55)	8.7 (6.0–11.0)	28 boys, 27 girls	52	3	0	0	55	10.0 (7.0–16.4)
Patients following HCT (*n* = 32)	8.0 (4.0–14.0)	17 boys, 15 girls	8	14	10	32	32	23 (12–34)
*p*-value	0.234	0.798	<0.001	<0.001	<0.001	<0.001	<0.001	<0.001

Abbreviations: ALL: acute lymphoblastic leukemia; AML: acute myeloid leukemia; HCT: hematopoietic cell transplantation; IQR: interquartile range; *n*: number of patients; PRBC: packed red blood cells; *p*-value: probability value.

**Table 2 cancers-17-02495-t002:** Serum concentrations of TMPRSS6, NEO1, and sHJV in healthy children and pediatric oncology patients across treatment stages.

Laboratory Parameter	Healthy Pediatric Group (*n* = 19)	Newly Diagnosed Patients (*n* = 43)	Patients Following Chemotherapy (*n* = 55)	Patients Following HCT (*n* = 32)	*p*-Values (vs. Healthy)
TMPRSS6 [ng/mL], median (IQR)	0.463 (0.382–0.545)	0.545 (0.421–0.837)	0.550 (0.455–0.731)	0.647 (0.449–0.750)	chemotherapy: *p* = 0.04
					HCT: *p* = 0.008
NEO1 [ng/mL], median (IQR)	0.696 (0.448–0.862)	0.855 (0.463–1.579)	0.881 (0.631–1.477)	0.987 (0.542–1.352)	chemotherapy: *p* = 0.02
					HCT: *p* = 0.03
sHJV [ng/mL], median (IQR)	63.47 (55.86–75.15)	45.09 (38.29–57.25)	45.93 (37.79–57.95)	38.98 (32.28–49.98)	newly diagnosed: *p* < 0.001
					chemotherapy: *p* < 0.001
					HCT: *p* < 0.001

Abbreviations: HCT: hematopoietic cell transplantation; IQR: interquartile range; NEO1: neogenin 1; sHJV: soluble hemojuvelin; TMPRSS6: matriptase-2.

**Table 3 cancers-17-02495-t003:** Comparative analysis of serum TMPRSS6, NEO1, and sHJV concentrations according to systemic iron availability, transfusional exposure, and inflammatory status.

Laboratory Parameter	Ferritin (Median: 708.3 µg/L)			PRBC Units (Median: 7)			CRP (Median: 0.87 mg/L)		
	Below Median	Above Median	*p*	Below Median	Above Median	*p*	Below Median	Above Median	*p*
TMPRSS6 [ng/mL], median (IQR)	0.498 0.388–0.672	0.630 0.454–0.780	0.003	0.508 0.394–0.716	0.596 0.454–0.748	0.027	0.554 0.458–0.746	0.534 0.420–0.732	0.567
NEO1 [ng/mL], median (IQR)	0.72 0.46–1.04	1.02 0.56–1.62	0.007	0.74 0.48–1.20	0.98 0.54–1.52	0.027	0.80 0.48–1.42	0.96 0.52–1.38	0.289
sHJV [ng/mL], median (IQR)	51 39–66	44 34–51	0.002	53 40–66	42 34–50	<0.001	48 40–60	43 34–57	0.045

Abbreviations: CRP: C-reactive protein; sHJV: soluble hemojuvelin; NEO1: neogenin 1; TMPRSS6: matriptase-2; *p*: significance in Mann–Whitney test; PRBC: packed red blood cells; IQR: interquartile ranges.

## Data Availability

The data presented in this study are available on request from the corresponding author due to privacy restrictions.

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
