# Peer review of "Iron Dysregulation Signature in Pediatric Leukemia: In-Depth Biomarkers of Iron Metabolism Involving Matriptase-2 and Neogenin-1"

_cancers, 2025, doi:10.3390/cancers17152495_

Round 1
Reviewer 1 Report
Comments and Suggestions for Authors
The study investigates the dysregulation of the TMPRSS6-NEO1-HJV axis in pediatric leukemia, providing novel insights into iron metabolism disturbances caused by transfusion-related iron overload. Given the high prevalence of transfusions in pediatric oncology and the lack of prior clinical data on these biomarkers, this work has significant implications for understanding iron-related complications and prognostic stratification.
While the study is scientifically sound and addresses an important gap, I recommend Major Revision due to several critical concerns that must be addressed before acceptance:
- In statistical analysis, clarify whether multiple-testing corrections were applied to correlation and subgroup analyses to avoid Type I errors.
- Expand beyond CRP (e.g., IL-6) to strengthen claims about inflammation’s role in sHJV suppression.
- Perform mechanistic validation and Include data on hepatic TMPRSS6 expression or furin activity to support serum biomarker interpretations.
- Justify pooling chemotherapy and HCT cohorts despite differing treatment intensities/outcomes.
- Perform multivariable Cox regression to confirm sHJV’s prognostic value independent of ferritin/PRBC units.
Comments on the Quality of English Language
Fine to me
Author Response
REPLY TO REVIEWER #1 COMMENTS
We would like to express our sincere gratitude to Reviewer 1 for the thorough and insightful review, which significantly contributed to the improved and more in-depth preparation of our manuscript.
- In statistical analysis, clarify whether multiple-testing corrections were applied to correlation and subgroup analyses to avoid Type I errors.
Reply: We sincerely thank the Reviewer for this valuable remark. In response, we have revised the Materials and Methods section to explicitly state that the Bonferroni-adjusted alpha threshold was applied in instances of multiple hypothesis testing within the same dataset. Importantly, implementation of this correction did not alter the statistical significance of our results, thereby confirming the robustness of the observed associations.
- Expand beyond CRP (e.g., IL-6) to strengthen claims about inflammation’s role in sHJV suppression.
Reply: We would like to sincerely thank Reviewer 1 for the thoughtful suggestion to expand beyond CRP and consider additional inflammatory markers such as interleukin-6 to strengthen the interpretation of sHJV suppression in the context of inflammation. We fully agree that this approach would provide a more comprehensive understanding of the immunometabolic mechanisms involved.
Unfortunately, at the current stage, we are unable to conduct further analyses due to two limiting factors. First, the research grant that supported our study has already been fully utilized, and additional funding is not presently available. Second, our remaining biological material is insufficient to perform the proposed cytokine assays in our cohort of nearly 150 pediatric patients. We sincerely apologize for these limitations. Nevertheless, we truly appreciate this inspiring idea and believe it may serve as a valuable foundation for a follow-up study in the future. Considering the importance of the Reviewer’s insightful suggestion, we have added the lack of more specific inflammatory markers as a limitation of our study in the revised Discussion section.
In addition, we acknowledge that it would be worthwhile to consider a broader spectrum of inflammatory biomarkers-potentially even more sensitive and specific than IL-6 or CRP. This is particularly relevant given the well-documented and strong association between circulating IL-6 and CRP concentrations (Brain Behav Immun. 2018;70:61-75), which may limit their individual discriminative value in certain inflammatory contexts.
As an alternative, we would like to note that we have access to procalcitonin concentrations in a subset of patients. However, this marker is primarily used for the diagnosis of bacterial infections and may not reliably reflect the state of systemic, non-infectious inflammation relevant to our study.
Once again, we thank the Reviewer for their insightful and constructive feedback. Please be assured that we made every effort to explore the available options and to present the most robust analysis possible within the scope and resources of the present study. We hope this suggestion will serve as a starting point for future research expanding on our current findings.
- Perform mechanistic validation and Include data on hepatic TMPRSS6 expression or furin activity to support serum biomarker interpretations.
Reply: We fully acknowledge the Reviewer’s valuable recommendation regarding the mechanistic validation of serum biomarker data through hepatic TMPRSS6 expression or furin activity analysis. While this would undoubtedly provide deeper biological insight, it presents substantial practical and ethical challenges in a clinical setting-particularly in pediatric patients. Assessing hepatic gene expression or protease activity would require invasive procedures, such as liver biopsy, which cannot be justified in the absence of direct clinical indications.
Nonetheless, we agree that mechanistic studies-possibly in experimental or preclinical models-could substantially advance the understanding of the TMPRSS6-NEO1-sHJV axis. We hope to explore these directions in future research.
We sincerely thank the Reviewer for this insightful suggestion, which we have explicitly acknowledged and addressed among the limitations discussed in the revised Discussion section.
- Justify pooling chemotherapy and HCT cohorts despite differing treatment intensities/outcomes.
Reply: We sincerely thank the Reviewer for this important comment. A similar concern was raised by Reviewer 2, who recommended that we refrain from pooling these two clinically distinct cohorts. In response to both Reviewers’ remarks and after careful clinical reconsideration of the pediatric subgroups, we have revised the manuscript accordingly. In the updated version, the analyses are now presented separately for patients treated with chemotherapy and those who underwent hematopoietic cell transplantation (HCT), without combining these groups (new Figure 2). We are grateful for this valuable suggestion, which has helped us improve the clarity and rigor of our study. Accordingly, we have revised both the Discussion and the Conclusions sections to accurately reflect the results of the updated subgroup analyses and to ensure consistency with the absence of statistically significant associations.
- Perform multivariable Cox regression to confirm sHJV’s prognostic value independent of ferritin/PRBC units.
Reply: We thank the Reviewer for this pertinent and thoughtful comment. In response to the suggestion, we conducted additional statistical analyses using the Cox proportional hazards regression model to evaluate the independent prognostic significance of soluble hemojuvelin (sHJV), adjusted for potential confounding variables such as serum ferritin concentration and the cumulative number of transfused packed red blood cell (PRBC) units. This analysis was performed across three defined cohorts: patients post-hematopoietic cell transplantation (HCT), those treated with chemotherapy, and a combined HCT+chemotherapy group.
In the univariate analyses, sHJV emerged as a statistically significant prognostic factor exclusively within the combined HCT+chemotherapy group, with associations observed for both overall survival (OS; p = 0.045) and event-free survival (EFS; p = 0.042). However, within the isolated HCT and chemotherapy subgroups, no variable, including sHJV, demonstrated statistical significance. Consequently, multivariable Cox regression analyses were not pursued, as the assumptions of model parsimony and the absence of statistically relevant predictors were not fulfilled.
In light of these findings, and given that the results do not expand upon or substantively modify the conclusions drawn from our primary analyses, we have opted not to include them in the revised manuscript. Furthermore, based on a critical appraisal of the existing literature and prevailing statistical methodology, the application of Cox proportional hazards modeling in the absence of statistically significant separation in Kaplan-Meier survival curves remains methodologically debatable. Without evidence of hazard divergence, the interpretive value of multivariable survival modeling becomes limited and may yield spurious inferences. Therefore, in alignment with statistical rigor and to preserve the clarity of the manuscript, these exploratory results are not presented in the final version.
We are grateful to the Reviewer for raising this point, which allowed us to reflect more deeply on the analytical approach and its limitations.
Reviewer 2 Report
Comments and Suggestions for Authors
I have read the paper analyzing levels of iron metabolism paramters in pediatric AL patients and controlos with great interest. Here are my comments:
1) number of patients in specific subgroups should be stated in the abstract (newly diagnosed AL, patients post-chemotherapy, patients following hematopoietic cell transplantation, healthy controls)
2) authors elaborate that OS and EFS were primary endpoints of the study, however, considering data presentation it is obvious that the analysis was cross-sectional in nature and that probably primary interest was to evaluate concentrations of specific iron metabolism paramters in relationship to therapeutic stage. Thus please correct this if you agree.
3) in my opinion the approach to analysis of OS and EFS is inherently biased by the way it is currently performed. I base this on several observations: The first observation is that the study was underpowered regarding specific subgroup sizes to detect survival assocations in the study. The second observation is that meaningful prognostic conclusions can be obtained only in the treatment naive patients since the timing of evaluation of parameter concentrations differed from clinically meaningful timepoints, and these parameters are highly affected by a high number of phenomena (inflammation, nutrition, infections, transfusions...). The third observation is that investigated subgroups are highly heterogenous regarding contexts, and probably underlying confounders affecting the measurements, thus I find it innapropriate to conjoin them into common analyses, even regarding two subgroups as currently attempted. Thus I advise to provide only exploratory analysis regarding survival for specific subgroups and delete the conjoined analyses. Also critically evaluate limitations of such approach.
Author Response
REPLY TO REVIEWER #2 COMMENTS
We would like to sincerely thank Reviewer 2 for the kind, thorough, and highly professional review. We truly appreciate the constructive feedback and valuable suggestions, which have helped us improve the clarity and overall quality of the manuscript. All comments have been carefully addressed, and the manuscript has been revised accordingly.
- number of patients in specific subgroups should be stated in the abstract (newly diagnosed AL, patients post-chemotherapy, patients following hematopoietic cell transplantation, healthy controls)
Reply: We thank the Reviewer for this valuable comment. In accordance with the suggestion, we have added the number of patients in each subgroup (newly diagnosed AL, post-chemotherapy, post-HCT, and healthy controls) to the revised version of the abstract.
- authors elaborate that OS and EFS were primary endpoints of the study, however, considering data presentation it is obvious that the analysis was cross-sectional in nature and that probably primary interest was to evaluate concentrations of specific iron metabolism paramters in relationship to therapeutic stage. Thus please correct this if you agree.
Reply: We thank the Reviewer for this important observation. We agree that the primary objective of our study was to assess iron metabolism parameters in relation to the therapeutic stage of pediatric acute leukemia, rather than to evaluate survival outcomes as primary endpoints. Accordingly, we have revised the relevant sections of the manuscript to clarify that overall survival (OS) and event-free survival (EFS) were included as exploratory analyses. We appreciate this comment, which helped us refine the focus and structure of the manuscript.
- in my opinion the approach to analysis of OS and EFS is inherently biased by the way it is currently performed. I base this on several observations: The first observation is that the study was underpowered regarding specific subgroup sizes to detect survival associations in the study. The second observation is that meaningful prognostic conclusions can be obtained only in the treatment naive patients since the timing of evaluation of parameter concentrations differed from clinically meaningful timepoints, and these parameters are highly affected by a high number of phenomena (inflammation, nutrition, infections, transfusions...). The third observation is that investigated subgroups are highly heterogenous regarding contexts, and probably underlying confounders affecting the measurements, thus I find it inappropriate to conjoin them into common analyses, even regarding two subgroups as currently attempted. Thus I advise to provide only exploratory analysis regarding survival for specific subgroups and delete the conjoined analyses. Also critically evaluate limitations of such approach.
Reply: We sincerely thank the Reviewer for this thorough and insightful evaluation. We fully agree with the concerns raised regarding the limitations inherent in our survival analyses. As correctly noted, the sample sizes of the specific subgroups were limited, the timing of biomarker assessment did not coincide with standardized clinical milestones, and the heterogeneity of patient subgroups introduced potential confounding factors. In light of these important considerations, we have removed all pooled survival analyses and now present only exploratory analyses within the chemotherapy and HCT subgroups separately. Moreover, the Discussion and Limitations sections have been revised to critically reflect the methodological constraints and the strictly exploratory nature of the survival data. We are grateful for this valuable guidance, which has substantially improved the clarity and scientific integrity of the manuscript.
Round 2
Reviewer 1 Report
Comments and Suggestions for Authors
The authors Addressed my comments very carefully.
Reviewer 2 Report
Comments and Suggestions for Authors
Thank You, the revision is satisfactory.